# Investigating the Smuts: Common Cues, Signaling Pathways, and the Role of *MAT* in Dimorphic Switching and Pathogenesis

**DOI:** 10.3390/jof6040368

**Published:** 2020-12-16

**Authors:** Teeratas Kijpornyongpan, M. Catherine Aime

**Affiliations:** Department of Botany and Plant Pathology, Purdue University, West Lafayette, IN 47907, USA or tkijporn@purdue.edu

**Keywords:** smut fungi, mating locus, Tween40, molecular signaling pathway, filamentous growth, yeast growth, morphological transition, comparative genomics

## Abstract

The corn smut fungus *Ustilago maydis* serves as a model species for studying fungal dimorphism and its role in phytopathogenic development. The pathogen has two growth phases: a saprobic yeast phase and a pathogenic filamentous phase. Dimorphic transition of *U. maydis* involves complex processes of signal perception, mating, and cellular reprogramming. Recent advances in improvement of reference genomes, high-throughput sequencing and molecular genetics studies have been expanding research in this field. However, the biology of other non-model species is frequently overlooked. This leads to uncertainty regarding how much of what is known in *U. maydis* is applicable to other dimorphic fungi. In this review, we will discuss dimorphic fungi in the aspects of physiology, reproductive biology, genomics, and molecular genetics. We also perform comparative analyses between *U. maydis* and other fungi in Ustilaginomycotina, the subphylum to which *U. maydis* belongs. We find that lipid/hydrophobicity is a potential common cue for dimorphic transition in plant-associated dimorphic fungi. However, genomic profiles alone are not adequate to explain dimorphism across different fungi.

## 1. Introduction

Fungi is one of the most diverse eukaryotic kingdoms of which members have colonized nearly every habitat and evolved multiple nutritional modes and strategies for survival and reproduction. In terms of growth and development, there are two common types of growth in fungi: filamentous growth and yeast growth. Most described fungi have an ability to grow as tube-like filamentous cells termed ‘hypha’, which serve as a basic unit for complex multicellular structures such as vegetative mycelia, dormant structures like sclerotia, asexual spore-bearing structures, and the sometimes complex fruiting bodies that produce sexual spores [1,2]. Some species, however, can survive, grow, and reproduce as a solitary cell, termed ‘yeast’ [3]. A subset of fungi has evolved the ability to grow both as unicellular yeasts and as multicellular hyphae (or, occasionally, as chains of incomplete bud cells termed pseudohyphae). These are called the ‘dimorphic fungi’.

Dimorphic fungi are predominantly found in Dikarya (a subkingdom including Ascomycota and Basidiomycota), although this phenomenon was originally described in the zygomycete genus *Mucor* ([4], Table 1). In Ascomycota, the best known of these are the ‘thermally dimorphic fungi’—many of which are also known as ‘black fungi’ or ‘black yeasts’ [5]. These fungi can be pathogenic to humans, and temperature serves as a stimulus for morphological transition between the hyphal and yeast phases [6]. Despite this shared character, thermally dimorphic fungi are a polyphyletic group found in three different ascomycete classes—Sordariomycetes, Eurotiomycetes and Dothideomycetes (Table 1). Another well-known dimorphic fungus is *Candida albicans* (Saccharomycotina), which is an opportunistic human pathogen. Dimorphic plant pathogenic fungi appear to differ from the thermally dimorphic fungi in not requiring temperature changes to induce different growth phases. Examples of dimorphic plant pathogens include the ascomycetes *Taphrina deformans* (Taphrinomycotina), which causes peach leaf curl disease, *Ophiostoma ulmi* and *O. novo-ulmi* (Sordariomycetes), which cause Dutch elm disease and *Zymoseptoria tritici* (Dothideomycetes), which causes Septoria leaf blotch on wheat. Recent studies have shown that the early-diverging basidiomycete lineages Pucciniomycotina, Ustilaginomycotina and Tremellomycetes also comprise many dimorphic fungi [7,8,9]. However, in most instances these receive less attention as most are neither economically important nor detrimentally harmful to humans. The best studied of the basidiomycete dimorphic fungi is the model species *Ustilago maydis,* which causes a disease in corn known as corn smut (Ustilaginomycotina, Basidiomycota).

*Ustilago maydis* is a saprobic yeast in its haploid stage, which can be grown in axenic cultures but is rarely detected in nature [7]. To undergo morphological transition, *U. maydis* requires mating from a genetically compatible partner, determined by mating (*MAT*) loci, as well as host signal perception [10]. After that, the pathogen transitions to its dikaryotic (having two haploid nuclei per cell) phase that includes both a switch to filamentous growth and a trophic switch from saprobic to pathogenic *in planta*. The pathogen ends its infectious cycle by transforming its hyphal masses into teliospores, which serve as both the dormant and reproductive propagules. Finally, meiosis occurs within teliospores that germinate to produce yeast cells for a new cycle. 

Recent research has focused on how *U. maydis* undergoes dimorphic transition and becomes pathogenic to a host. However, studies in other non-model dimorphic fungi are limited and very few genetic tools/resources are available for these. The development of advancing technologies such as high-throughput sequencing and more rapid methods for genome assembly and annotation can provide reference data for non-model species to facilitate comparative studies [11,12,13]. 

Here, we provide a comprehensive review, combined with data from pilot studies, of (1) physiological factors affecting fungal dimorphism, (2) the mating system contributing to fungal dimorphism, and (3) underlying molecular mechanisms of dimorphic transition. With the extensive body of data available for *U. maydis* as a base, we make multiple comparative analyses between the *U. maydis* system and that of other phytopathogenic dimorphic fungi, drawn from preliminary physiological and genomic studies of other Ustilaginomycotina species. For the purpose of this review, we inclusively consider any fungus that has both a yeast phase and a filamentous phase, consisting of either true hyphae or pseudohyphae, as a dimorphic fungus. 

## 2. Physiological Aspect of Fungal Dimorphism

### 2.1. External Factors Affecting Fungal Dimorphism

Because most dimorphic fungi can be grown in axenic culture, there is a body of physiological studies to provide an overview of known triggers in dimorphic transition (Table 1). Nitrogen starvation is widely known to trigger yeast-to-filamentous growth in several species such as *Saccharomyces cerevisiae*, *Schizosaccharomyces pombe*, *Candida albicans*, *Cryptococcus neoformans*, *Ustilago maydis* and *Zymoseptoria tritici* [14,15,16,17,18,19]. In contrast, *Trichosporon cutaneum* and *Yarrowia lipolytica* represent exceptions in which high nitrogen promotes filamentous growth [20,21]. The type of nitrogen source also affects transition in several dimorphic species [20,22,23,24,25,26]. The effect of carbon source on fungal dimorphism has been reported, but less frequently. For instance, fermentable hexoses can promote yeast growth in several *Mucor* species [27], while *S. cerevisiae* undergoes invasive filamentous growth when cultured in the absence of fermentable sugars [28]. The sugar derivative N-acetyl glucosamine serves as an elicitor for filamentous growth of *C. albicans*, *Y. lipolytica*, *Histoplasma capsulatum* and *Blastomyces dermatidis* [16,29,30].

Other environmental factors can also impact dimorphic switching. Acidic pH induces filamentous growth in two basidiomycete dimorphic fungi—*U. maydis* and *T. cutaneum* [20,31]; Pseudohyphal growth is induced in *Tremella fusiformis* under pH 5–7 [25]; *Y. lipolytica* and *C. albicans* form hyphae under alkaline pH [24,32]. It has long been known that anaerobic condition, or high CO_2_ tension, triggers yeast growth of several *Mucor* species [27]. More recently, these have been found to promote hyphal growth in at least two ascomycete dimorphic fungi, *C. albicans* and *Y. lipolytica* [16,24]. Interestingly, there is a single report showing an effect of Zn^2+^ on promoting filamentous growth in *Aureobasidium pullulans* [33]. Finally, physical properties such as temperature, media solidity and agitation can affect growth form in culture of some dimorphic fungi [20,34,35].

**Table 1 jof-06-00368-t001:** Examples of dimorphic fungi, life history strategies, and environmental cues for morphological transition.

Lineages/Species	Life Strategy	Environmental Cues *	References
ASCOMYCOTA Saccharomycotina *Saccharomyces cerevisiae*	Saprobe	Nutrient limitation (carbon, nitrogen)	[14]
*Candida albicans*	Opportunistic human pathogen	Temperature, serum, CO_2_, pH, farnesol, GlcNAc	[16]
*Holleya sinecauda*	Plant pathogen	Media solidity	[34]
*Yarrowia lipolytica*	Saprobe	Nitrogen source, GlcNAc, serum, citrate, pH, anaerobic	[21,24,29]
Taphrinomycotina *Taphrina deformans*	Plant pathogen	Unknown cue from host leaves	[36,37]
*Schizosaccharomyces pombe*	Saprobe	Nitrogen starvation	[15]
Eurotiomycetes *Blastomyces dermatidis*	Human pathogen	Temperature, GlcNAc	[6,30]
*Coccidioides immitis*	Human pathogen	Temperature	[6]
*Talaromyces marneffii*	Human pathogen	Temperature	[6]
*Histoplasma capsulatum*	Human pathogen	Temperature, GlcNAc	[6,30,38]
Sordariomycetes *Sporothrix schenckii*	Human pathogen	Temperature	[6]
*Ophiostoma ulmi* & *O. novo-ulmi*	Plant pathogen	Nitrogen source, inoculum density, quorum-sensing, linoleic acid	[23,39,40,41]
*Verticillium albo-atrum*	Plant pathogen	Culture agitation, inoculum density	[35]
*Metarhizium rileyi*	Insect pathogen	Host hemolymph, quorum-sensing	[42]
Dothideomycetes *Zymoseptoria tritici*	Plant pathogen	Nitrogen starvation	[19]
*Aureobasidium pullulans*	Saprobe	Nitrogen source, quorum-sensing, Zn^2+^	[22,33]
*Hortaea werneckii* **	Saprobe, opportunistic human pathogen	Temperature, CO_2_, cysteine, inoculum size, agitation	[43,44]
BASIDIOMYCOTA Ustilaginomycotina *Ustilago maydis*	Plant pathogen	Mating, lipid, hydrophobicity, acidic pH, nitrogen starvation	[18,31,45,46]
*Malassezia* spp.	Opportunistic human pathogen	L-DOPA, lipid on mammal skin, high CO_2_ tension	[47,48,49]
Pucciniomycotina *Microbotryum lychnidis-dioicae*	Plant pathogen	Unknown	[50]
Agaricomycotina *Cryptococcus neoformans*	Human pathogen	Mating, nitrogen starvation, temperature, CO_2_	[17]
*Trichosporon cutaneum*	Saprobe	Nitrogen source, pH, temperature	[20]
*Tremella* spp.	Saprobe, mycoparasite	Mating, ploidy status, carbon sources, nitrogen sources	[25,26]
MUCOROMYCOTA *Mucor* spp.	Saprobe, opportunistic human pathogen	Carbon source, CO_2_	[27]

* GlcNAc = N-acetyl glucosamine, L-DOPA = L-3,4-dihydroxyphenylalanine. ** The genus is also reported in the literature as *Cladosporium* and *Exophiala.*

In addition to environmental cues, dimorphic pathogens also utilize signals from their hosts to undergo morphological transition as well as to initiate infection and colonization. Temperature serves as a primary factor for triggering dimorphic transition in many human pathogenic fungi (Table 1). Most of these are adapted to grow as yeasts at 37 °C (human body temperature), and as hyphae in a cooler environment [6,17,44]. High CO_2_ tension, which is presumably found in human tissues, can also promote yeast growth in some dimorphic pathogens that cause systemic infection [17,27,44]. Conversely, human body temperature, high CO_2_ and blood serum are factors that promote filamentous growth in *C. albicans* [16]. The commensal and opportunistic fungus *Malassezia* is likely to require signals on mammal skins for filamentous growth, which is enhanced under a microaerophilic environment [49,51]. While signals involved in fungal dimorphism have been extensively studied in human pathogens, very little data exist for switching mechanisms in phytopathogenic fungi. In the well-studied dimorphic plant pathogen *U. maydis*, acid pH, lipid and hydrophobicity have been demonstrated to promote filamentous growth [31,46,52]. Linoleic acid can also induce hyphal growth in the Dutch-elm disease pathogen *O. novo-ulmi* [41]. These cues are commonly found on the plant host surface [53], suggesting that the pathogens perceive the host for dimorphic transition. The peach leaf curl pathogen *Taphrina deformans* has filamentous growth only when it colonizes a host, possibly after perceiving an unknown signal [37].

### 2.2. Comparative Physiological Studies in Ustilaginomycotina

We conducted a pilot experiment in *U. maydis* and closely related species in Ustilaginomycotina to determine what signals might serve as a common cue for plant-associated dimorphic fungi [12]. By varying several types of carbon sources, we found that the lipid-mimic Tween40 triggers dimorphic transition in most dimorphic Ustilaginomycotina species (Figure 1 and Figure 2). These confirm findings from prior work demonstrating that plant oils, fatty acids and Tween40 promote filamentous growth of *U. maydis* [54]. More recent studies also reveal the effect of hydroxy-fatty acids and hydrophobic surfaces on promoting hyphal growth of the *U. maydis* solopathogenic strain SG200 [46]. Of the Ustilaginomycotina species we sampled, those exhibiting filamentous growth under Tween40 as a carbon source are all phylloplane fungi (i.e., found on plant leaf surfaces)—*U. maydis* (Ustma), *Jaminaea rosea* (Jamro), *Meira miltonrushii* (Meimi), *Violaceomyces palustris* (Viopa) and *Tilletiopsis washingtonensis* (Tilwa) [55,56,57,58]. It is possible that they utilize lipids/hydrophobicity as signals upon arrival on a host surface. Hydrophobicity and lipid compounds are also involved in morphogenesis in other plant pathogenic fungi, such as appressoria formation in the rice blast fungus *Magnaporthe oryzae* and the powdery mildew fungus *Blumeria graminis* [59,60] and yeast-to-hyphal transition in *O. ulmi* and *O. novo-ulmi* [40,41]. Because the abovementioned fungi interact with a wide variety of plant host species, lipids and hydrophobicity may be a commonly utilized signal to trigger dimorphic transition in plant-associated fungi in general.

However, Tween40 does not promote hyphal growth in other studied species such as the smut fungus *Testicularia cyperi* and the phylloplane fungus *Moesziomyces aphidis* (Figure 1 and Figure 2) and it is evident that different fungi sense different host signal molecules. For example, *Magnaporthe oryzae* can sense surface hydrophobicity, cutin monomers or leaf waxes to trigger appressoria formation [59]. As the environment on plant surfaces is rich in lipids but poor in nitrogen compounds [53], some plant-associated fungi such as *S. cerevisiae* and *Z. tritici* may perceive nitrogen starvation as a signal that they have arrived on host surfaces [19,61]. Other dimorphic fungi may perceive other biomolecules inside plant hosts. For instance, we found that pectin, a common component of plant cell walls, triggers yeast-to-hyphal transition in two dimorphic fungi—*V. palustris* and *T. cyperi* ([12]; Figure 1 and Figure 2), the first such study to implicate pectin in phase transitions. More intriguingly, our study shows temperature as a trigger for dimorphic transition in some Ustilaginomycotina species such as *U. maydis*, *Moesziomyces aphidis*, *T. cyperi* and *V. palustris* ([12], Figure 1 and Figure 2). However, none of the energy-source carbohydrates tested in our study significantly promotes hyphal growth in our studied species. Finally, we found three Ustilaginomycotina species with consistent monomorphic growth, regardless of treatment. *Acaromyces ingoldii* has constitutively hyphal growth, whereas *Pseudomicrostroma glucosiphilum* and *Meira* sp. MCA4637 have constitutively yeast growth ([12]; Figure 1 and Figure 2).

## 3. Mating and Fungal Dimorphism

### 3.1. Cellular Communication in Dimorphic Fungi

Intercellular communication is another critical factor for determining growth phases in many dimorphic fungi. Inoculum density plays a major role in dimorphism of several ascomycete species such as *Ophiostoma ulmi* and *O. novo-ulmi*, *Verticillium albo-atrum*, *Metarhizium rileyi* and *Aureobasidium pullulans* [22,35,39], suggesting quorum-sensing activity in these fungi. Farnesol, as a renowned quorum-sensing molecule, is secreted by *C. albicans* to suppress its filamentous growth and biofilm formation [16]. Other quorum-sensing molecules involved in fungal growth and development are discussed in a recent review [62]. Although quorum-sensing is poorly known in basidiomycete fungi, mating is associated with dimorphism in many species. Hyphal growth is often found in the post-mating dikaryotic stage of dimorphic fungi in Pucciniomycotina, Ustilaginomycotina and Tremellomycetes [63]. These include two model species: the human pathogenic fungus *Cryptococcus neoformans* [17] and the corn smut fungus *U. maydis* [64]. Here we focus on the plant pathogen *U. maydis*; a review of the mating system of *C. neoformans* and its role in filamentous growth can be found elsewhere [16,64].

### 3.2. Mating (MAT) Loci in U. maydis

In smut fungi such as *U. maydis*, the mating process is critical for dimorphic transition from yeast to filamentous growth, as well as for switching to a pathogenic phase for host infection. This has been demonstrated through *MAT* deletion studies in *U. maydis* [64,65,66], *Sporisorium reilianum* [67], and *S. scitamineum* [68]. There are two *MAT* loci in smut fungi: *MAT a* and *MAT b*. The *MAT a* locus comprises a pheromone precursor gene (*Mfa*) and a pheromone receptor gene (*Pra*). An expressed pheromone precursor protein is cleaved and then coupled with a farnesyl group to make a functional lipopeptide pheromone [69]. Then, the released pheromone binds to a pheromone receptor from another yeast cell that has a different *MAT a* allele. This binding leads to the pheromone response, cell cycle arrest at the G2 stage, and conjugation tube formation [10]. Reciprocal pheromone perception of two mating-compatible yeast cells results in conjugation tube formation and cytoplasmic fusion to become dikaryotic. The *MAT b* locus comprises genes encoding homeodomain transcription factors *bE* and *bW*. These transcription factors become active once *bE* from one *MAT b* allele heterodimerizes with *bW* from another *MAT b* allele, and vice-versa. The bE/bW heterodimer then regulates downstream genes that are involved in filamentous growth and pathogenic development. However, a cell cycle does not resume until the dikaryotic fungus perceives a signal from a plant host and successfully penetrates plant tissues [10]. In a few rare cases, some smut fungi can undergo pathogenic development without prior mating, termed solopathogens. Solopathogenic strains can originate either from diploid strains [70,71], or via genetic modification to artificially generate a single haploid strain with compatible *bE* and *bW* genes that can form a self-heterodimer [72].

In order to be genetically compatible for mating, both haploid yeast cells need to have different alleles at both the *MAT a* and *b* loci. When both *MAT* loci are unlinked, a number of mating types are possible as different combinations of the *MAT a* alleles and *MAT b* alleles. A fertilized diploid zygote can bear offspring with four possible mating types, termed a ‘tetrapolar’ mating system. In *U. maydis* and *S. reilianum*, there are up to three alleles of the *MAT a* locus and up to 25 alleles of the *MAT b* locus [69]. This allelic diversity would increase the chances for outcrossing with other individuals within a population [69,73]. Other smut fungi such as *U. hordei* and *S. scitamineum* have linked *MAT a* and *b* loci. Therefore, a fertilized diploid zygote can bear offspring with only two possible mating types, termed a ‘bipolar’ mating system. The bipolar mating system decreases the odds for outcrossing compatibility due to dependent assortment of both *MAT* loci, while promoting selfing/inbreeding as there is a 50% chance of compatibility with offspring in the same cohort, but only a 25% chance for sibling mating in the tetrapolar mating system [69,73]. A recent study of Kellner et al. [74] revealed that there are three conserved alleles in the *MAT a* locus among smut fungi in Ustilaginales, regardless of whether the *MAT* arrangement is bipolar or tetrapolar. The allelic conservation of the *MAT a* locus is also demonstrated by interspecific pheromone response [74]. Therefore, we postulate that genetic diversity at the *MAT b* locus is a primary determinant for reproductive isolation across different species, as well as for increasing intraspecific outcrossing compatibility in the tetrapolar mating system.

### 3.3. Comparative MAT Loci Analyses in Ustilaginomycotina

We sampled sixteen representative genomes in Ustilaginomycotina to examine how the arrangement of the *MAT* loci is conserved in this subphylum ([13,75,76,77,78,79,80,81]; Figure 3). First, we found that most studied species have the tetrapolar mating system, except *Testicularia cyperi* and *Malassezia globosa,* which are bipolar, and *Exobasidium vaccinii,* which is strictly bipolar (i.e., genes in the *MAT b* locus are located right next to genes in the *MAT a* locus). We found that gene arrangement near the *MAT a* and *b* loci is conserved in Ustilaginales, including *U. maydis*, *Pseudozyma hubeiensis*, *S. reilianum*, *P. antartctica*, and *T. cyperi*. In *Violaceomyces palustris*, a few neighboring genes located next to the *bW/bE* genes are syntenic to those in other examined Ustilaginales species. A gene encoding a DEAD-box helicase is proximal to the *bW/bE* genes for most studied Ustilaginomycotina species, except *Jaminaea rosea*, *Pseudomicrostroma glucosiphilum*, *Acaromyces ingoldii*, *E. vaccinii*, *Meira miltonrushii* and *V. palustris*. In addition, bW or bE protein sequences are truncated in *Testicularia cyperi*, *Tilletiaria anomala*, *P. antartctica*, *M. globosa* and *M. sympodialis* (Figure 3, Appendix A). As most of these species are found to be dimorphic, this truncation may only affect the mating process but not fungal dimorphism. The truncation of the *bE* genes in *Malassezia* may explain why mating has never been observed for this fungus despite the presence of both the *MAT a* and *b* loci [69].

For the *MAT a* locus, gene synteny is conserved across species in Ustilaginales (Figure 3). Although conservation in gene synteny in other Ustilaginomycotina species is not in evidence, two genes encoding the proteasome regulatory subunit Rpn10 and the mRNA deadenylase CAF1 are frequently linked to a pheromone receptor gene. A pheromone precursor gene is not found in *A. ingoldii* and *P. antarctica*. It is notable that *A. ingoldii* is a constitutively filamentous fungus. The gene phylogeny of pheromone receptor genes reveals that most sampled genomes possess either the *MAT a1* or *a2* allele (Appendix A). Considering the low prevalence of the *a3* allele in Kellner et al. [74] and in our study, we hypothesize that the *a3* allele has recently emerged in Ustilaginales.

In contrast to the conservation found within Ustilaginales, our analyses show a remarkable diversification of *MAT* arrangement in Exobasidiales species, including *A. ingoldii*, *E. vaccinii* and *M. miltonrushii*. No pheromone precursor gene is identified in *A. ingoldii*; a strictly bipolar *MAT* arrangement with a duplicate of the pheromone precursor gene and the pheromone receptor gene is found in *E. vaccinii*; and a duplicate of the *bE* gene and the pheromone receptor gene is found in both the *MAT* a and *b* loci of *M. miltonrushii* (Figure 3). The duplication of the *MAT* genes serves as another strategy for increasing mating type compatibility, which is commonly found in mushroom-forming fungi [69]. In short, the arrangement of the *MAT* loci in Ustilaginomycotina is much more diversified than previously known and is dependent on data from non-model species to fully appreciate this. Nonetheless, organization of the mating system alone cannot explain diversity of fungal growth forms in Ustilaginomycotina. As mating and ploidy status are typically coupled with fungal dimorphism of many early-diverging basidiomycetes (Ustilaginomycotina, Pucciniomycotina and Tremellomycetes) [63], it is interesting to further investigate the molecular mechanism of this coupling, as well as what causes decoupling in monomorphic species.

## 4. Molecular Mechanism Related to Fungal Dimorphism

After signal perception, dimorphic fungi require an elaborate system of signal transduction, gene regulation and biomolecular localization for morphogenetic reprogramming. Upstream signaling pathways are the most intensively studied as they are widely conserved. In fungi, there are two major conserved pathways—the cAMP/Protein kinase A (PKA) pathway and the mitogen-activated kinase (MAPK) pathway [82,83]. In this review, we will provide a comprehensive overview of these signaling pathways, as well as any upstream and downstream molecular players, using *U. maydis* as a primary model (summarized in Figure 4).

### 4.1. cAMP-PKA Pathway

The cyclic-AMP/protein kinase A (cAMP-PKA) pathway is a well conserved signaling pathway across eukaryotic organisms [83]. The signaling initiates by the perception of an extracellular stimulus through a G protein-coupled receptor (GPCR). Once a ligand binds to the receptor, it interacts with a heterotrimeric G protein complex, which is bound to the receptor, to activate a G_α_ subunit by incorporating a GTP molecule. Then, the active G_α_ subunit dissociates from the G protein complex and subsequently activates an enzyme called adenylate cyclase. This enzyme catalyzes a conversion of ATP to 3’,5’ cyclic AMP (cAMP). The cAMP serves as a second messenger that amplifies a signal to downstream targets. One of the most important targets is protein kinase A (PKA), a hetero-tetrameric enzyme constituting two regulatory subunits and two catalytic subunits. The cAMP molecule binds to the regulatory subunits, causing them to release from the catalytic subunits. The PKA catalytic subunits can then phosphorylate many downstream proteins involved in various cellular responses. Finally, to provide a feedback control, a phosphodiesterase enzyme reduces an intracellular cAMP level by hydrolyzing a phosphodiester bond of cAMP, yielding AMP as a product.

Most components of the fungal cAMP-PKA pathway have been characterized in *U. maydis*. Regenfelder et al. [84] discovered four G_α_ genes (*Gpa1*–*Gpa4*). Only *Gpa3* shows a distinct phenotype from the others. The *gpa1, gpa2* and *gpa4* single mutants result in constitutively yeast growth and no response to pheromone, while the *gpa3* mutant has a promoted filamentous growth but does not respond to a pheromone. A subsequent discovery of G_β_ (*Bpp1*) revealed a similar biological function to *Gpa3* [85]. *Uac1* was identified as an adenylate cyclase in *U. maydis* [86], for which the mutant has constitutively filamentous growth. Although a hetero-tetrameric structure of PKA is yet to be validated, the regulatory subunit *Ubc1* and the catalytic subunits *Adr1* and *Uka1* have been identified. The *ubc1* mutant has a defect in filamentous growth *in vitro* [86], wherein it can colonize a host with hyphae but cannot undergo gall formation during the dikaryotic stage [87]. The *adr1* mutant is constitutively filamentous in the haploid stage, but its pathogenicity is defective [88]. The other PKA catalytic subunit *Uka1* does not show any significant effect on growth form and pathogenicity [88]. Finally, Agarwal et al. [89] recently characterized two phosphodiesterase genes *Umpde1* and *Umpde2*. The deletion of *umpde1* results in significantly reduced pathogenicity. While the *umpde2* deletion mutant has a subtle effect on pathogenicity, its filamentous growth is strongly attenuated.

Overall, several molecular genetic studies have shown that the activation of the cAMP-PKA pathway promotes budding but reduces filamentous growth of *U. maydis in vitro*. This trend is confirmed by the addition of exogenous cAMP, which suppresses filamentous growth [86,90,91]. Meanwhile, filamentous growth is a consequence of the pathway activation in *C. albicans* and *S. cerevisiae* [14,16]. Defects of several components in the pathway often lead to hampered pathogenic development in *U. maydis*, suggesting that the cAMP-PKA pathway plays multiple roles in both cellular and pathogenic developments. In addition, dimorphic transition to filamentous growth alone is not sufficient for the switching from a saprobic phase to a pathogenic phase.

### 4.2. MAPK Pathway

The mitogen-activated protein kinase (MAPK) pathway is another extensively studied pathway in many eukaryotic systems. The pathway is activated through a sequential phosphorylation of Serine/Threonine/Tyrosine protein kinases, termed the MAPK cascade. There are three major groups of kinases in the MAPK cascade. A MAPK kinase kinase (MAPKKK) is activated by an upstream molecule such as a G-protein complex, a GTPase, or another kinase. The active MAPKKK phosphorylates a MAPK kinase (MAPKK) to enable its activity. After that, the MAPKK phosphorylates a MAPK to trigger its function as a final step in the cascade. Through phosphorylation, the activated MAPK can either stimulate or inhibit many downstream targets such as protein kinases, protein phosphatases, or transcription factors.

Components of the MAPK cascade have been described in *U. maydis*—*Kpp4*/*Ubc4* as a MAPKKK gene [92,93], *Fuz7*/*Ubc5* as a MAPKK gene [93,94] and *Kpp2*/*Ubc3*, *Kpp6* and *Crk1* as MAPK genes [95,96,97,98]. In addition, two genes directly involved in the cascade have been identified. *Ubc2* encodes a pheromone-responsive adaptor protein that physically interacts with Kpp4 [99,100]. The other gene is a dual specificity phosphatase named *Rok1*, which negatively controls Kpp2 and Kpp6 through dephosphorylation [101].

Most known genes in the MAPK cascade of *U. maydis* are involved in pheromone response, conjugation tube formation, filamentous growth and pathogenic development [92,93,94,95,96,98,102]. An exception is *Kpp6*, of which the mutant has a normal morphogenesis but attenuated pathogenicity [97]. The Kpp4-Fuz7-Kpp2 MAPK cascade, corresponding to the Ste11-Ste7-Kss1 cascade in *S. cerevisiae*, affects dimorphic transition to filamentous growth [14,82]. A Kss1 paralog named Fus3, another downstream target of Ste7, has a separate role in *S. cerevisiae* mating [14]. However, in *U. maydis* the mating function is complemented by Kpp2, while its paralog Kpp6 has a single function in pathogenic development. Crk1 in *U. maydis* is another downstream target of Fuz7. Crk1 function, having a role in filamentous growth, mating and pathogenesis [98,102], can also be regulated through phosphorylation by Kpp2 and cAMP-mediated mechanism. Ime2, an ortholog of Crk1 in *S. cerevisiae*, is involved in pseudohyphal growth and meiotic progression via a cAMP-mediated pathway, although its upstream MAPK counterpart is still unknown [103]. These examples demonstrate the functional diversification of the MAPK pathway, as well as its potential crosstalk with the cAMP/PKA pathway. *Saccharomyces cerevisiae* has at least two other MAPK cascades—the Bck1-MKK1/2-Sit2 cascade involved in cell wall integrity and the Ste11/Ssk2/Ssk22-Pbs2-Hog1 cascade involved in high osmolarity response [82]. Interestingly, the osmotic sensor protein Sho1 in *U. maydis* can directly interact with Kpp6 to trigger appressorium formation on a hydrophobic surface [47], but whether Sho1 can interact with other components in the MAPK cascade remains unknown. Although recent comparative genomics reveals that almost all genes in other MAPK cascades have respective orthologs in *U. maydis* [82], none of these have been characterized. Therefore, it is interesting to investigate the other MAPK members to understand how they affect *U. maydis* growth and development.

The multidomain adaptor protein Ubc2 provides another example of functional diversification within a single molecule. Klosterman et al. [99] demonstrated that the SAM and RA domains at the N-terminus region of Ubc2 are essential for *U. maydis* filamentous growth. They suggested a direct interaction of the SAM domain and Kpp4 as an underlying mechanism to activate the MAPK cascade. Two C-terminal SH3 domains of Ubc2 are dispensable for filamentous growth but required for pathogenesis [99]. Other proteins that interact with Ubc2 to trigger a different response are yet to be characterized. To summarize, the *U. maydis* MAPK pathway frequently has a coupled function for both filamentous growth and pathogenic development.

### 4.3. Alternative Pathways

In addition to the cAMP/PKA and MAPK pathways, GTPases are important signaling components for filamentous growth. A few families of GTPases have been described in *U. maydis*: Ras, Rac and Rho families. There are two described genes in the Ras family, *Ras1* and *Ras2*. *Ras2*, acting upstream of the MAPK cascade, contributes to filamentous growth, mating response, and pathogenicity [104]. A subsequent study showed that the *Ras2* function is activated by the guanyl nucleotide exchange factor Sql2 [105]. While *Sql2* overexpression leads to constitutively filamentous growth, its deletion mutant has normal mating but adversely affects pathogenic development [105]. *Ras1*, less studied than *Ras2*, is only known to promote the expression of the pheromone gene [105]. Rac is another GTPase family known for polarized growth and cell wall expansion in *U. maydis* [106]. Overexpression of *Rac1* causes *U. maydis* to display filamentous growth during the haploid stage, while overexpression of a hyperactivated form of Rac1 leads to an isotropic cell wall expansion and cell lethality [106]. Rac1 is activated by the guanyl nucleotide exchange factor Cdc24, but potentially inhibited by the Rho1 GTPase [107]. The only known downstream target of Rac1 is the PAK family kinase Cla4, which modulates polarized growth, cell separation and bud neck formation [107,108]. The cyclin-dependent kinase Cdk5 is also involved in cell polarity through modulating Cdc24 localization at a growing tip [109].

A few other signaling pathways related to fungal dimorphism have been reported in other dimorphic fungi. One is the two-component signaling system, which is thoroughly investigated in thermally dimorphic fungi [110]. The pathway is activated by signal perception through a hybrid histidine kinase, which induces phosphorylation of its response regulator at a histidine residue. After that, a histidine phosphotransfer protein mobilizes a phosphate group from the response regulator to a downstream target to activate/inhibit its function. Recent studies have shown that the class III hybrid histidine kinase plays a crucial role for dimorphic transition in *Histoplasma capsulatum*, *Talaromyces marneffii* and *Blastomyces dermatidis* [110]. Another pathway is the calcium signaling pathway. Upon a signal perception that leads to calcium ion influx, the elevated intracellular Ca^2+^ activates a calcium-binding protein called calmodulin. The Ca^2+^-bound calmodulin then interacts with other downstream proteins to trigger their functions. One of these is a Serine/Threonine protein phosphatase named calcineurin, which is found to control dimorphic transition in *Mucor circinelloides* and *Paracoccidioides brasiliensis* [111,112]. In *M. circinelloides*, calcineurin also contributes to spore size regulation, cell wall integrity and antifungal drug resistance [113]. A single study of *U. maydis* identifies the catalytic subunit of calcineurin Ucn1 as an antagonist of PKA [114].

### 4.4. Upstream and Downstream Molecular Players

Signal perception is one of the least studied areas in signal transduction for several reasons. One reason is that most receptors are transmembrane proteins, which are difficult for studying protein structures and interactions. Another concerns the exhaustive process of matching which ligands bind to which receptors. In *U. maydis*, the only confirmed ligand-receptor interaction is the lipopeptide pheromone and the seven-transmembrane pheromone receptor Pra1 [64]. The pheromone binding results in the activation of the MAPK cascade [92,93,100]. The high-affinity ammonium transporter Ump2 has recently been found to promote filamentous growth of *U. maydis* under low-ammonium conditions [18,115]. An interaction between Ump2 and Rho1 suggests a mechanism for filamentous growth via the Rho1-Rac1 signaling pathway. This suggests that Ump2 acts as a receptor for nitrogen starvation in addition to a transporter function. Sho1 and Msb2 are other two characterized transmembrane proteins that interact with the MAPK cascade [52]. These two proteins contribute to stress response and pseudohyphal growth in *S. cerevisiae*, whereas they play a role in appressoria formation but not filamentous growth in *U. maydis* [52]. Other signals, such as acidic pH and lipids, are involved in *U. maydis* filamentous growth. However, their corresponding receptors are yet to be discovered.

The pheromone response factor 1 (Prf1) is a master transcription factor that controls the mating process, filamentous growth and pathogenic development of *U. maydis* through the activation of genes in the *MAT* loci [10,45]. Many studies have investigated how *Prf1* itself is regulated. The promoter region of *Prf1* is relatively long with a length of at least 2 kilobases [116]. Prf1 can autoactivate itself by binding to its promoter to initiate transcription [116,117]. The promoter region is also subject to several transcription factors such as Rop1, Pac2, Hap2 and Med1 [116,118,119], and the regulation networks of these genes have been investigated [118,120]. A recent study reveals that the histone deacetylase Hos2 facilitates Prf1 transcription through unwinding the Prf1 open-reading frame region from histone molecules [121]. In addition to transcriptional control, Prf1 activity is also regulated via post-translational phosphorylation. Prf1 has multiple Serine/Threonine sites that are subject to phosphorylation by PKA and Kpp2 [122]. The PKA phosphorylation sites are required to induce gene expression in both *MAT a* and *b* loci, while the MAPK phosphorylation sites are essential for only *MAT b* locus expression. Therefore, Prf1 represents one of a few known molecules that integrate signaling from different pathways and fine-tune downstream cellular responses.

Once the dikaryotic stage is formed, the bE/bW heterodimer can transcriptionally regulate dozens of genes [123]. Heimel et al. [124] revealed that the bE/bW heterodimer may not directly regulate these downstream targets but does so via another master regulator called Rbf1. Several genes in pathogenesis and cell cycle regulation have been identified as targets of Rbf1 [124]. Therefore, the bE/bW regulatory cascade is crucial for filamentation and pathogenic development during the dikaryotic stage. After *U. maydis* fully colonizes a host plant, it undergoes a transition from vegetative growth to teliospore formation. A recent study has identified the WOPR family transcriptional factor Ros1 as a primary regulator for this switching [125]. One of Ros1 functions includes a feedback inhibition of the bE/bW regulatory cascade.

Although Prf1 serves as a central regulator for dimorphic transition, there are a few cases wherein *U. maydis* undergoes filamentous growth in a Prf1-independent manner. For instance, Lee and Kronstad [104] demonstrated that hyperactivated Ras2 can lead to a constitutively filamentous growth in *prf1* mutant. The *prf1* mutant can also form hyphae under induction by acidic pH or lipids [54,91]. Accordingly, there must be other downstream molecular players that have yet to be identified. Hgl1, a direct phosphorylation target of PKA that contributes to *U. maydis* filamentous growth [126], is so far not known to interact with Prf1. The GATA transcription factor Nit2 has recently been identified as a response factor for filamentous growth under nitrogen catabolite repression [127]. The *nit2* deletion mutant does not alter expression of *Prf1* and its upstream genes, suggesting that *Nit2* may be either downstream or independent from *Prf1*. The histone acetyltransferase Gcn5 also contributes to *U. maydis* dimorphism [128], but whether its function is Prf1-dependent remains unknown. To summarize, there are several alternative pathways that can induce dimorphic transition in *U. maydis*. However, the Prf1-mediated regulatory pathway is still crucial for pathogenic development of *U. maydis* as it leads to mating, dikaryotization and the establishment of the bE/bW heterodimer, which subsequently turns on many downstream effectors and pathogenesis proteins.

After sequences of signal transduction and transcriptional regulation, target genes for cell structures and morphogenesis are ultimately activated. In *U. maydis* these include cytoskeletal motors such as the class V myosin gene *Myo5* and the kinesin genes *Kin1* and *Kin3* [129,130,131]. Chitin synthases are another group of genes that affect fungal dimorphism. Weber et al. [132] found that the polar chitin synthases Chs5 and Chs7 are required for conjugation tube formation, dikaryotic hyphae formation and pathogenicity, while the Myosin-V chitin synthase Mcs1 only influences polarized growth once entrance to plant tissues has been achieved. In addition, cell division genes such as the B-type cyclin gene *Clb2*, the nuclei distribution gene *Clp1*, and the septin gene *Sep3* are involved in *U. maydis* filamentous growth [133,134,135]. *Yup1* encodes an endosomal *t*-SNARE protein that is important for exocytosis and endocytosis; this is found to be critical for polarized growth [136]. The cell end markers Tea1 and Tea4, indicating a site for polarized growth, have been recently characterized in *U. maydis* [137,138]. Finally, the RNA-binding proteins Khd4 and Rrm4 have been demonstrated to control regular filamentous growth [139,140]. Rrm4 functions in long-distance mRNA transport so that transcripts involved in polarized growth can be expressed at a precise location [10,141]. Khd4 recognizes a certain sequence motif in RNA, suggesting a role in post-transcriptional regulation [140].

### 4.5. Comparative Genomics of Fungal Dimorphism Genes in Ustilaginomycotina

We performed comparative analyses to examine how genes involved in fungal dimorphism are conserved across different species in Ustilaginomycotina [13,75,76,77,78,79,81,142]. We found that genes in the cAMP/PKA pathway, the MAPK pathway, and the GTPase-mediated signaling are highly conserved in all studied species, except *Jaminaea rosea*, *Malassezia globosa*, *Moesziomyces aphidis*, *Pseudozyma antarctica*. *Pseudozyma hubeiensis*, *Pseudomicrostroma glucosiphilum* and *Tilletiopsis washingtonensis,* of which only 1–2 genes are absent (Table 2). The pheromone receptor gene *Pra1* and the high-affinity ammonium transporter gene *Ump2* are found in all species, suggesting conserved machinery for signal perception.

Surprisingly, *Malassezia globosa* and *Moesziomyces aphidis* are the two species with the greatest number of absent genes. *Prf1* and its regulator genes such as *Med1* and *Rop1* are not found in the *Malassezia globosa* genome (Table 2). Moreover, a few downstream transcription factors of the bE/bW cascade—*Biz1*, *Clp1* and *Ros1*—are also not found. This multiple gene loss may be associated with a drastic shift of *Malassezia* ecological niches from plant-associated fungi to animal-associated fungi [13]. In *Moesziomyces aphidis*, several polar tip-associated genes are lost from its genome, including the chitin synthase genes *Chs7* and *Mcs1*, the RNA-binding protein gene *Rrm4* and the cell end marker genes *Tea1* and *Tea4* (Table 2). Although *Moesziomyces aphidis* is able to display filamentous growth, hyphal production is limited compared to other dimorphic fungi [143,144]; our study. 

## 5. Benefits of Fungal Dimorphism

It has been hypothesized that dimorphic fungi utilize different growth forms to maximize fitness in different stages of their life cycle. For instance, filamentous growth in *Saccharomyces cerevisiae*, as a chain of elongated cells under nutrient-limited condition, is considered as a scavenging response to increase the chances of finding nutrients [14]. In many dimorphic human pathogens, the switch to unicellular yeast growth includes a drastic shift in cell wall composition to avoid recognition by host immune cells [110,143]. Recent reviews also suggested that the yeast form is beneficial for dissemination through the host bloodstream and respiratory tracts, and for thermotolerance when living inside the human body [51,110,143]. However, the opportunistic pathogen *C. albicans* resides on a host’s skin and mucosal layers as a commensal yeast, while the hyphal form can penetrate host tissues and evade host immunity [16,144]. *Moesziomyces aphidis*, one of the Ustilaginomycotina species that exhibits promoted filamentous growth at high temperatures (Figure 1 and Figure 2), has recently been found as the hyphal growth form in immunocompromised patients [145,146]. The entomopathogenic fungus *Metarhizium rileyi* switches to yeast-like growth upon entering the hemolymph of an insect. This helps the fungus spread throughout the host body. Once reaching a threshold critical density, it switches back, possibly by quorum-sensing, to the hyphal growth form to complete colonization and kill the host [42]. Many basidiomycete mycoparasites utilize the hyphal growth to penetrate host tissues and form a specialized structure called haustoria to absorb nutrients from the host [63]. However, their ecological roles during the saprobic yeast phase remains poorly studied.

For dimorphic plant pathogens, filamentous growth is a primary form that penetrates the epidermis and colonizes host tissues [147]. Genetic studies in *U. maydis* have shown that mutants with reduced filamentous growth often have attenuated virulence [66,98,102,127,134]. Although *Tilletiopsis washingtonensis* is a common yeast-like fungus typically recovered from leaf phylloplanes, it has recently been found producing extensive hyphal growth on apple skins, causing the postharvest disorder ‘white haze’ syndrome [58,148]. On the other hand, the yeast form can be beneficial for dissemination through plant vascular systems as shown in *Ophiostoma* and *Verticillium* [147]. Moreover, it is tempting to propose that the yeast form is more advantageous for passive dispersal through wind, rain or animal vectors. For example, Comeau et al. [149] suggested that the yeast form of *Ophiostoma novo-ulmi* is involved in passive dispersal of yeast-like budding spores to other elm trees through bark beetle vectors. 

## 6. Conclusions and Perspectives

Fungal dimorphism is a sophisticated process that involves signal perception from multiple sources (environmental cues, cellular communication and host signals), reticulation of signal transduction (the cAMP/PKA pathway, the MAPK pathway and the GTPase-mediated pathway) and a complex regulatory network of gene expression. As the most extensively studied species, the corn smut fungus *Ustilago maydis* still serves as an excellent model for investigating fungal dimorphism and its role in pathogenic development. However, examination of non-model species is critical to understand which mechanisms are conserved and which are species-specific responses. For example, lipids and hydrophobicity, as common plant surface cues, appear to trigger yeast-to-hyphal transition in many plant-associated dimorphic fungi, with few exceptions. In addition, we show that data from mating gene studies and comparative genomics may not be enough to explain the difference in growth forms of these Ustilaginomycotina species. Integration of various approaches such as collection of more natural isolates, physiological screening and comparative transcriptomics are necessary for better understanding of this complex biological phenomenon.

According to our data, dimorphic transition is not an all-or-none response. For instance, some treatments promote filamentous growth of dimorphic fungi, but the proportion of filamentous cells is still lower than 50% (Figure 2). Several studies include only a few photomicrographs as a representation of growth form [20,54,150]. As the growth forms of dimorphic fungi are dynamic both in terms of cell population and time [20,40,42], we strongly encourage the use of quantitative approaches, combined with statistical analyses, to support the findings as well as to determine the strength of phenotypes. Protocols used to distinguish yeast and filamentous cells should be clearly defined as these can be critical for result interpretation. Detailed methods have been described in systems like *Candida albicans* and *Ophiostoma* [39,41,151].

Defining ‘dimorphic fungi’ is often a challenging task, especially for less widely known species. One major reason is a lack of adequate observation. Many mycologists describe species based on their primary growth form observed in nature or in axenic culture. However, as dimorphic fungi may switch to another growth form only under certain conditions, they may be erroneously classified as either yeasts or filamentous fungi. For example, *Moesziomyces aphidis* was originally described as a yeast species with the presence of elongated cells resembling pseudohyphae [152], but recent studies show that it can grow as short hyphae [146,153]. Moreover, distinguishing between true hyphae and pseudohyphae is somewhat difficult unless their cytology is carefully investigated. Hence, an inclusive term ‘filamentous growth’ is often used to avoid any confusion. Finally, clear definitions for terms should be established. Some researchers define dimorphism as any yeast-like fungus that can undergo filamentous growth [154,155], some limit the definition to only yeast-like fungi that form true hyphae as opposed to pseudohyphae [143], and some do not provide a clear distinction [156]. A new term, ‘polymorphic fungi’, has recently been applied for any fungi that can grow as yeasts, pseudohyphae and true hyphae [143,156]. Establishing clear definitions of terminology would facilitate the expansion of research in this area.

## 7. Materials and Methods

### 7.1. Physiological Study

The following strains were included in this study: *Ustilago maydis* TKC58 ≡ NRRL Y-64004, *Moeziomyces aphidis* MCA6183 ≡ NRRL Y-64005, *Testicularia cyperi* MCA3645 ≡ ATCC MYA-4640, *Violaceomyces palustris* SA807 ≡ CBS 139708, *Meira miltonrushii* MCA3882 ≡ ATCC-MYA 4883 ≡ CBS 12591, *Meira* sp. MCA4637 ≡ NRRL 66775, *Acaromyces ingoldii* MCA4198 ≡ CBS 140884, *Tilletiopsis washingtonensis* MCA4186 ≡ NRRL Y-63783, *Jaminaea rosea* MCA5214 ≡ CBS 14051 and *Pseudomicrostroma glucosiphilum* MCA4718 ≡ CBS14053.

We utilized Holliday’s minimal liquid media [71] as the base media for the study. Carbon sources with 1% concentration (*w*/*v*) were varied according to different treatments—glucose as a standard carbon source used in laboratory, sucrose as a representative for a mobilized carbon source in plants, soluble starch as a representative for a storage carbon source in plants, pectin as a representative for a structural carbohydrate of plant cell walls, and Tween40 as a mimic of the waxy, hydrophobic surface of plants. For high temperature (35 °C) treatments we used 1% glucose (*w*/*v*) as a carbon source. To set up an experiment, we cultured each fungal strain in yeast malt broth for 3 days. After that, cell suspensions were centrifuged and rinsed three times in sterile water. The number of 1 × 10^6^ cells was inoculated into 5 mL of culture media, resulting in 2 × 10^5^ cells/mL as an initial cell concentration. Most inoculated cells appeared as yeast growth, except *A. ingoldii* which appear as filamentous growth. There were five replicates for each treatment. The cultures were observed after five days of incubation at room temperature (25 °C), except the high temperature treatment at 35 °C, and 180 rpm shaking condition.

To determine fungal growth forms, cell morphologies were observed under the BH-2 Olympus phase contrast microscope (Olympus Corp., Tokyo, Japan). For each sample, at least two hundred cells were counted and classified into two categories: yeast cells and filamentous cells. A cell bud was counted as one yeast cell, while an elongated fungal structure longer than two yeast cells without fungal cell wall constriction between cells was counted as filamentous cells. The Olympus SC30 camera was used to capture photomicrographs of fungal morphologies through the Olympus cellSens version 1.8 software. A percentage of filamentous cells was used as a proxy for types of growth form. We utilized R version 3.4.1 under RStudio version 1.0.153 for data analyses and visualization. Mean and standard deviation were reported for each treatment as a bar graph using the ggplot2 package [157]. One-way analysis of variance (ANOVA) was used to analyze difference in growth forms among treatments for each species; the Tukey’s HSD test was used as a post-hoc comparison. *P*-values less than 0.01 were considered statistically significant.

### 7.2. MAT Loci Studies

Sixteen Ustilaginomycotina genomes were used in these analyses: *Ustilago maydis* [75], *Sporisorium reilianum* [76], *Pseudozyma hubeiensis* [77], *Pseudozyma antarctica* [78], *Malassezia globosa* [79], *Malassezia sympodialis* [80], *Tilletiaria anomala* [81], *Exobasidium vaccinii* [158], *Acaromyces ingoldii*, *Ceraceosorus guamensis*, *Jaminaea rosea*, *Meira miltonrushii*, *Pseudomicrostroma glucosiphilum*, *Testicularia cyperi*, *Tilletiopsis washingtonensis* and *Violaceomyces palustris* [13]. Genes in *MAT* loci were determined by blasting reference protein sequences from *U. maydis* genes against other species. The reference genes from *U. maydis* are as follows: *Pra1* (UMAG_02383) for the *MAT* a locus, *bE* (*HD1*, UMAG_00577) and *bW* (*HD2*, UMAG_00578) for the *MAT* b locus. Once the *MAT* loci for each genome were identified, gene synteny was examined within 20 kilobases upstream and downstream of the mating genes. A pheromone precursor gene is sometimes unannotated due to its short gene length. Thus, we scanned with 20 kb upstream/downstream range from the pheromone receptor gene to confirm the presence/absence of the pheromone precursor gene using the FGENESH online portal with *U. maydis* as a reference (http://www.softberry.com/).

To analyze the *MAT* configuration within a phylogenetic context, protein sequences of the mating genes (pheromone receptor genes, *bE*, and *bW*) were used. The protein sequences of each gene were aligned using the MUSCLE algorithm performed in MEGA-X [159]. The alignments were visually inspected for any protein truncation. For the pheromone receptor genes, we incorporated additional reference protein sequences from *U. maydis* and *S. reilianum* to represent all three *MAT a* alleles. The sequences, with GenBank accessions, are as follows: CAI59749 for the *a1* allele in *S. reilianum* [67], P31303 for the *a2* allele in *U. maydis* [65] and CAI59763 for the *a3* allele in *S. reilianum* [67]. Finally, the aligned protein sequences of the pheromone receptor genes were used to reconstruct a phylogenetic tree through the Neighbor-joining method performed in MEGA-X with JTT as a protein substitution model and 1000-replicate bootstrapping as a node support value.

### 7.3. Comparative Genomic Studies

Sixteen Ustilaginomycotina genomes were used in the comparative genomic analyses. These include all genomes utilized in the mating gene studies, except *Malassezia sympodialis*, plus *Moesziomyces aphidis* [145]. We determined orthologous genes through OrthoFinder 2.1.2 [160] using protein models as inputs. Then, we selected only orthogroups containing *U. maydis* dimorphism genes (Appendix A). Presence/absence patterns were indicated by non-zero/zero values of orthologs in each species. For orthogroups that have paralogs, we inspected a gene tree to determine gene orthologs. Any genes belonging to the same monophyletic clade as a reference gene were considered as true orthologs.

## Figures and Tables

**Figure 1 jof-06-00368-f001:**
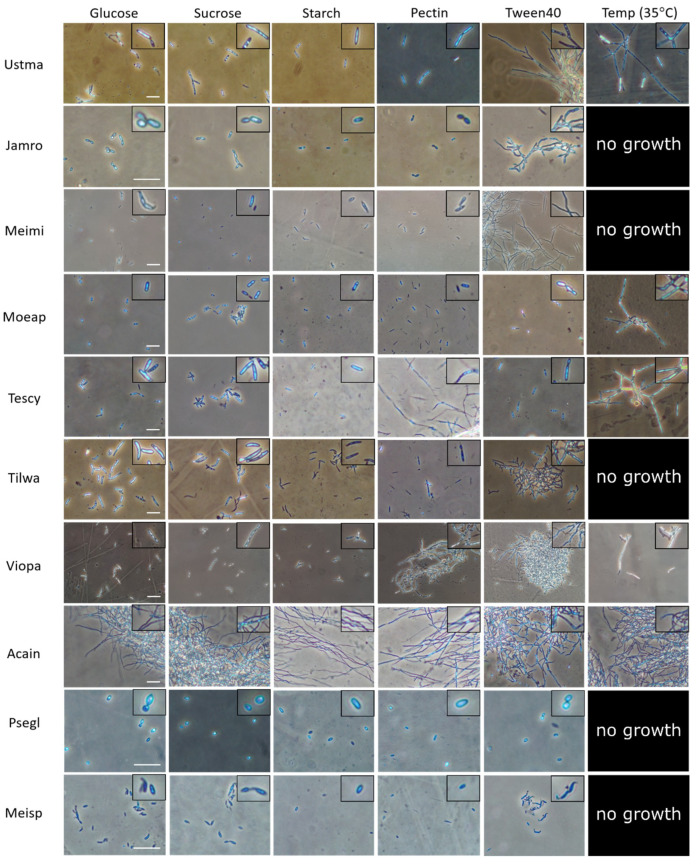
Effects of carbon sources and temperature on fungal dimorphism of Ustilaginomycotina. Cell morphologies of ten Ustilaginomycotina species were observed after treatment under different carbon sources (glucose, sucrose, starch, pectin, Tween40) and high temperature (35 °C) with glucose as a carbon source. Most treatments were incubated at 25 °C for 5 days. Fungal cultures were observed under phase contrast microscopy. An inlay photomicrograph in each cell is magnified up to 1.5 times to better visualize cell morphology. Species abbreviations are as follows: Ustma, *Ustilago maydis*; Jamro, *Jaminaea rosea*; Meimi, *Meira miltonrushii*; Moeap, *Moesziomyces aphidis*; Tescy, *Testicularia cyperi*; Tilwa, *Tilletiopsis washingtonensis*; Viopa, *Violaceomyces palustris*; Acain, *Acaromyces ingoldii*; Psegl, *Pseudomicrostroma glucosiphilum*, Meisp, *Meira* sp. MCA4637. Bars: 20 μm.

**Figure 2 jof-06-00368-f002:**
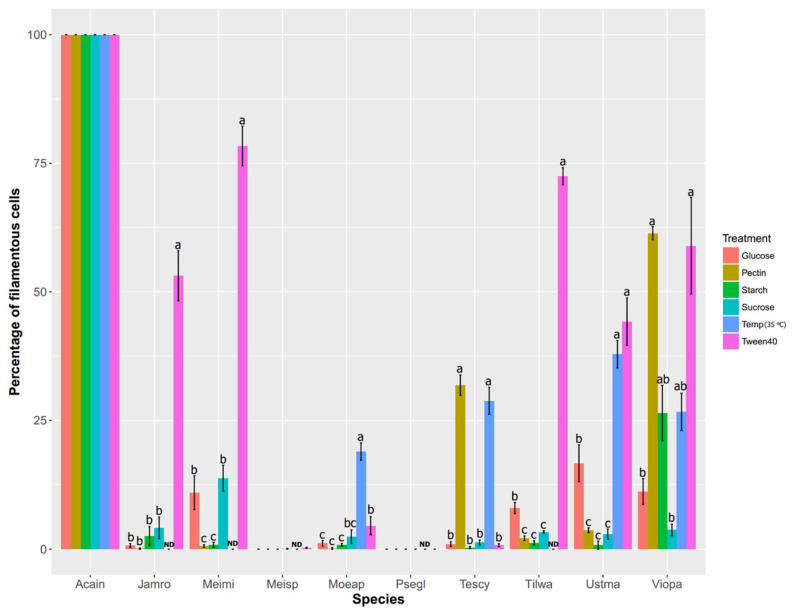
Growth form quantification of ten Ustilaginomycotina species supplied with different carbon sources (red, gold, green, aqua and pink) and high temperature (blue). This figure is a statistical representation of data from Figure 1. The growth form is reported as percentages of filamentous cells. One-way analysis of variance (ANOVA) was used to test differences among treatments for each species. Letters indicate statistical groups inferred from the Tukey HSD post-hoc test, with ‘a’ as the highest value, ‘b’ as an intermediate value and ‘c’ as the lowest value. ND: no data, meaning that fungi do not grow in those conditions. Species abbreviations are as indicated in the Figure 1 legend.

**Figure 3 jof-06-00368-f003:**
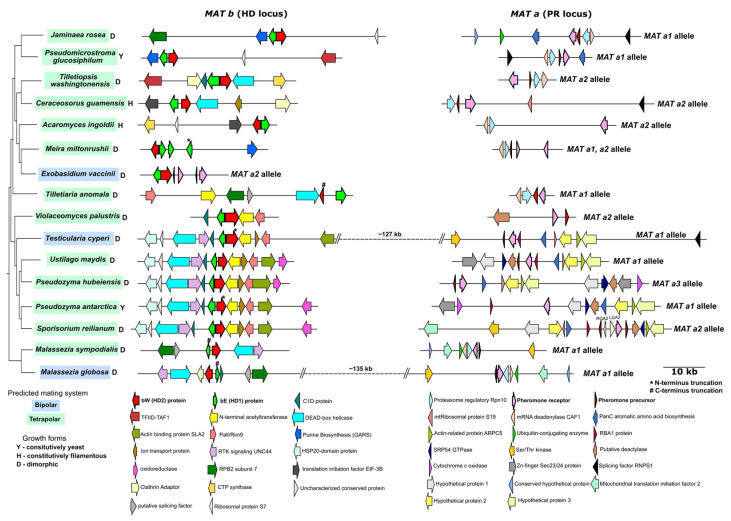
Comparative synteny of *MAT* loci in Ustilaginomycotina. Sixteen genomes of Ustilaginomycotina were incorporated in the analyses. Genes for pheromone receptor (*Pra*), bE and bW from *Ustilago maydis* were used as references to identify *MAT* loci in studied species. Gene synteny was examined within 20 kilobases upstream and downstream of the mating genes. Only genes that are syntenic in at least two species are shown. Gene orientation is indicated by arrowheads. Different arrowhead colors indicate different genes (see more details in the legend). Predicted mating system and growth form are indicated at each species’ name. Allele type of *MAT a* locus is determined by the gene phylogeny of pheromone receptor genes (Appendix A).

**Figure 4 jof-06-00368-f004:**
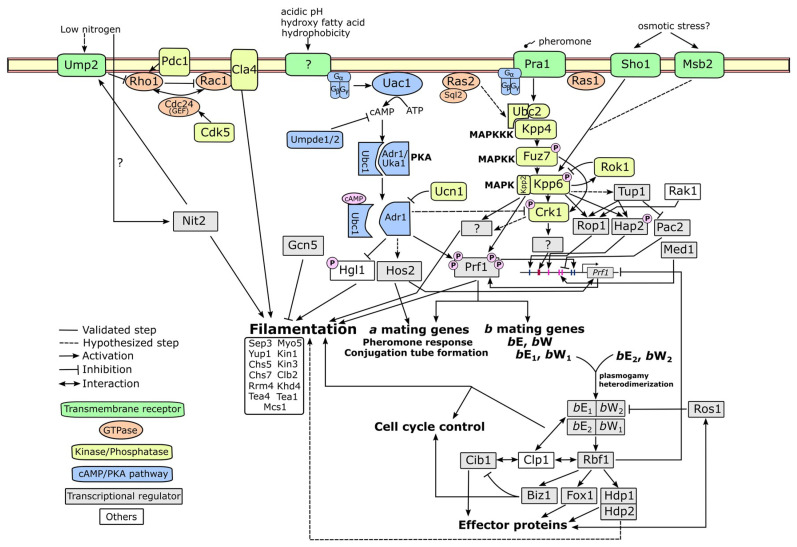
Overview of regulatory network for fungal dimorphism in *U. maydis*. The filamentous growth of *U. maydis* can be triggered through a few pathways: the cyclic-AMP/protein kinase A (cAMP/PKA) pathway, the mitogen-activated kinase (MAPK) pathway and the GTPase-mediated signaling pathway. These signal transductions trigger the alteration of transcriptional network that activates downstream effectors for morphogenesis. Details of molecular players in the figure can be found in Appendix A.

**Table 2 jof-06-00368-t002:** Comparative genomics of dimorphism genes across sixteen Ustilaginomycotina species. Colors in the left column indicate: green, receptor genes; blue, genes in the cAMP/PKA pathway; yellow, genes in the MAPK pathway; red, genes in the GTPase-mediated signaling; white, other downstream genes. Details about genes in this table can be found in Figure 4 and Appendix A. White and black cell colors indicate ortholog presence and absence, respectively. Species abbreviations are as follows: Acain, *Acaromyces ingoldii*; Cergu, *Ceraceosorus guamensis*; Exova, *Exobasidium vaccinii*; Jamro, *Jaminaea rosea*; Malgl, *Malassezia globosa*; Meimi, *Meira miltonrushii*; Moeap, *Moesziomyces aphidis*; Psean, *Pseudozyma antarctica*; Psehu, *Pseudozyma hubeiensis*; Psegl, *Pseudomicrostroma glucosiphilum*; Spore, *Sporisorium reilianum*; Tescy, *Testicularia cyperi*; Tilan, *Tilletiaria anomala*; Tilwa, *Tilletiopsis washingtonensis*; Ustma, *Ustilago maydis*; Viopa, *Violaceomyces palustris*.

Gene Name	Ustma GeneID	Acain	Cergu	Exova	Jamro	Malgl	Meimi	Moeap	Psean	Psehu	Psegl	Spore	Tescy	Tilan	Tilwa	Ustma	Viopa
*Msb2*	UMAG_00480																
*Pra1*	UMAG_02383																
*Sho1*	UMAG_03156																
*Ump2*	UMAG_05889																
*Adr1*	UMAG_04456																
*Bpp1*	UMAG_00703																
*Gpa3*	UMAG_04474																
*Uac1*	UMAG_05232																
*Ubc1*	UMAG_00525																
*Ucn1*	UMAG_00936																
*Uka1*	UMAG_11860																
*Umpde1*	UMAG_02531																
*Umpde2*	UMAG_10895																
*Crk1*	UMAG_11410																
*Fuz7/Ubc5*	UMAG_01514																
Kpp2/*Ubc3*	UMAG_03305																
*Kpp4/Ubc4*	UMAG_04258																
*Kpp6*	UMAG_02331																
*Rok1*	UMAG_03701																
*Ubc2*	UMAG_05261																
*Cla4*	UMAG_10145																
*Pdc1*	UMAG_01366																
*Ras1*	UMAG_01643																
*Ras2*	UMAG_03172																
*Rho1*	UMAG_05734																
*Sql2*	UMAG_10803																
*Biz1*	UMAG_02549																
*Chs5*	UMAG_10277																
*Chs7*	UMAG_05480																
*Cib1*	UMAG_11782																
*Clb2*	UMAG_10279																
*Clp1*	UMAG_02438																
*Gcn5*	UMAG_10190																
*Hap2*	UMAG_01597																
*Hgl1*	UMAG_11450																
*Hos2*	UMAG_11828																
*Khd4*	UMAG_03837																
*Kin1*	UMAG_04218																
*Kin3*	UMAG_06251																
*Mcs1*	UMAG_03204																
*Med1*	UMAG_03588																
*Myo5*	UMAG_04555																
*Nit2*	UMAG_10417																
*Pac2*	UMAG_15096																
*Prf1*	UMAG_02713																
*Rac1*	UMAG_00774																
*Rak1*	UMAG_10146																
*Rbf1*	UMAG_03167																
*Rop1*	UMAG_12033																
*Ros1*	UMAG_05853																
*Rrm4*	UMAG_03494																
*Sep3*	UMAG_03449																
*Tea1*	UMAG_15019																
*Tea4*	UMAG_01012																
*Tup1*	UMAG_03280																
*Yup1*	UMAG_05406

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
