# Peer review of "Investigating the Smuts: Common Cues, Signaling Pathways, and the Role of MAT in Dimorphic Switching and Pathogenesis"

_jof, 2020, doi:10.3390/jof6040368_

Round 1
Reviewer 1 Report
The manuscript by T Kijpornjongpan and MC Aime aims to review the dimorphism of Ustilago maydis in comparison with non-model dimorphic fungi. While this is partly achieved, it does consist in fact of three studies which are almost independent of each other. Also, not all three fit into the proposed scope of this manuscript as review paper. The first study is purely experimental, analyzing the efficiency of different triggers for dimorphism in several dimorphic Ustilaginales strains, and showing that in most cases the triggers only incompletely shift the ratio between yeast and hyphal cells. The second study is also not a review, but a comparative genome organization analysis of the MAT genes in several Ustilaginales, concluding that there seems no pattern related to dimorphism. The third is indeed mainly a review, namely on the signaling pathways with emphasis on their relevance for triggering the dimorphism switch in Ustilaginales, but is also features genome comparison.
The promise of the title is only partly fulfilled. The manuscript does quite some comparisons, but a discussion on the applicability of results obtained from the model for non-models, the degree of fitting for model species vs. non-model species, and on the validity of Ustilago as model species (e.g. for dimorphic fungi) is nowhere to be found. Perhaps the title might be rephrased to avoid disappointment in readers expecting these issues to be discussed.
In my opinion, the first and third study are generally really good manuscripts in itself, while the second might need some reframing. There the authors seem surprised at the lack of patterns explaining the different growth forms, and thus conclude that comparing the MAT gene organization does not help to understand dimorphism. I recommend to frame it differently. It is in no way surprising that organisms do not completely restructure sexual reproduction as one of their core functions for a shift of growth form. This is especially valid if, as it seems, the potential for dimorphism is present in all Ustilaginales, if not all basal basidiomycetes as seen with the Tremellomycetes.
Instead, I would assume that a mechanistic linkage between sexual development and growth form has evolved several times independently, and perhaps also been lost repeatedly, in accordance with the life strategy of the respective species. Finding out where and how this linkage occurred might then indeed be studied by checking overlap in the respective signaling pathways. If done in this way, the second and third study might be combined to a well readable single manuscript.
I do however not see a way to include the first study, since its scope and results seem not to supplement the other two studies. Since it is without doubt an interesting manuscript by itself, I recommend to publish it separately.
I also have a few more detailed comments and suggestions, in particular to chapter 2:
line 54ff: I am not certain that dimorphic basidiomycetes are understudied on behalf of low economic or medicinal impact, and also do not think Ustilago is the best studied of these. What about Cryptococcus, which is a human pathogen, at least similarly well studied and also dimorphic when diploid?
line 121ff: Ustilago is not the only well studied plant pathogenic dimorph. For example, there are reports on the dimorphism trigger of the plant pathogens Ceratocystis ulmi (Naruzawa et al. 2016: Botany 94:31-39; Nickerson et al. 2006: Appl. Environ. Microbiol. 72: 3805-3813), Mycosphaerella graminicola (Mehrabi et al. 2007: Mol. Plant Microbe In. 19: 1262-1269) and even something on Taphrina (Svetaz et al. 2017: Plant Cell Environ. 40: 1456-1473).
line 133ff: Fatty acids are also involved in dimorphism of Ceratocystis (Naruzawa et al. 2016: Botany 94:31-39).
line 161f: The growth of these three species is monomorphic only as far as your treatments could not trigger dimorphism. This does not imply that there is no dimorphism trigger at all.
line 207ff: MAT seems also involved in Cryptococcus dimorphism (Wickes 1996: Proc. Natl. Acad. Sci. USA: 93: 7327-7331), and probably also in Tremella, since only their dikaryons are filamentous (and also parasitic).
line 545ff: I do not think you mean the hyphal form is "used" to invade host cells, since that implies a user. Maybe it is just able to enter them.
References and figures are fine with me.
Table 2: I recommend to put table 2 to supplementary material. It disrupts the reading flow, and table 3 contains enough information to follow the text.
Table 3: Please indicate in the table legend what the blackened cells stand for.
Author Response
Point-by-point author response to reviewers
Reviewer 1
Comments and Suggestions for Authors
Point 1: The manuscript by T Kijpornjongpan and MC Aime aims to review the dimorphism of Ustilago maydis in comparison with non-model dimorphic fungi. While this is partly achieved, it does consist in fact of three studies which are almost independent of each other. Also, not all three fit into the proposed scope of this manuscript as review paper. The first study is purely experimental, analyzing the efficiency of different triggers for dimorphism in several dimorphic Ustilaginales strains, and showing that in most cases the triggers only incompletely shift the ratio between yeast and hyphal cells. The second study is also not a review, but a comparative genome organization analysis of the MAT genes in several Ustilaginales, concluding that there seems no pattern related to dimorphism. The third is indeed mainly a review, namely on the signaling pathways with emphasis on their relevance for triggering the dimorphism switch in Ustilaginales, but is also features genome comparison.
Response 1: Incorporating a few pieces of analyses to enhance discussion is not uncommon in recent review articles. See examples in Bakkeren et al. 2008. Fungal Genetics and Biology 45: S15 – S21; Riquelme and Bartnicki-Garcia 2008. Fungal Biology Reviews 22: 56-70; Kijpornyongpan et al. 2019. FEMS Yeast Research 19(2): foy125; Martinez-Soto et al. 2020 Microorganisms 8:1072; Libkind et al. 2020. FEMS Yeast Research 20: foaa042. However, most of them have parts of experiments/analyses embedded in the review, such as in a figure legend, and do not explicitly elaborate in the Materials and Methods section. We would like to explain what we have done in the Materials and Methods section to make sure other researchers can reproduce our work. Our article attempts to introduce each aspect of dimorphism by reviewing the literature, and then top up with our data to expand the border of research areas. Thus, our article may be considered as a hybrid between a research article and a review, but findings from the experiments/analyses are all used to enhance the discussion of the knowledge in the field.
Point 2: The promise of the title is only partly fulfilled. The manuscript does quite some comparisons, but a discussion on the applicability of results obtained from the model for non-models, the degree of fitting for model species vs. non-model species, and on the validity of Ustilago as model species (e.g. for dimorphic fungi) is nowhere to be found. Perhaps the title might be rephrased to avoid disappointment in readers expecting these issues to be discussed.
Response 2: We acknowledge the reviewer’s comment and rename the title as “Investigating the smuts: common cues, signaling pathways, and the role of MAT in dimorphic switching and pathogenesis”.
Point 3: In my opinion, the first and third study are generally really good manuscripts in itself, while the second might need some reframing. There the authors seem surprised at the lack of patterns explaining the different growth forms, and thus conclude that comparing the MAT gene organization does not help to understand dimorphism. I recommend to frame it differently. It is in no way surprising that organisms do not completely restructure sexual reproduction as one of their core functions for a shift of growth form. This is especially valid if, as it seems, the potential for dimorphism is present in all Ustilaginales, if not all basal basidiomycetes as seen with the Tremellomycetes.
Instead, I would assume that a mechanistic linkage between sexual development and growth form has evolved several times independently, and perhaps also been lost repeatedly, in accordance with the life strategy of the respective species. Finding out where and how this linkage occurred might then indeed be studied by checking overlap in the respective signaling pathways. If done in this way, the second and third study might be combined to a well readable single manuscript.
Response 3: Fungal dimorphism is typically found in early-diverging basidiomycetes including Ustilaginomycotina, Pucciniomycotina and Tremellomycetes. These dimorphic fungi normally have hyphal growth during a dikaryotic stage after mating (Boekhout et al. 2011 in The Yeasts: A taxonomic study Vol III 5th edition. Chapter 100). Therefore, it is convincing to assume that sexual development is coupled with their dimorphic transition. One hypothesis is that the mating genes activate downstream genes involved in dimorphic transition from yeast to filamentous growth, as demonstrated in U. maydis. According to our comparative analyses (Table 3 in the original version and Figure 3), the mating genes as well as other upstream signaling pathways (like the MAPK pathway, activated by pheromone recognition) are well conserved in Ustilaginomycotina species, no matter what type of growth forms they have. Therefore, we conclude that these presence/absence patterns alone could not explain growth forms in studied fungal species and there must be other unknown molecular players, potentially in downstream pathways, that contribute to the coupling of sexual development and fungal dimorphism. As the knowledge in this area is scarce, it is too cumbersome to conduct a study as requested by the reviewer.
Even though our study could not provide any evidence how mating process is coupled with dimorphic transition, we would like to provide an update on the mating system of U. maydis and allies as previous studies were published around 10 years ago (Bakkeren et al. 2008. Fungal Genetics and Biology 45: S15 – S21; Kellner et al. 2011. PLOS Genetics 7(12): e1002436). Incorporating additional species that do not belong to Ustilaginales can provide better understanding how the mating loci are conserved across the subphylum. We also mention about the coupling of mating and dimorphic transition in the first and the last paragraphs of the section 3 ‘Mating and fungal dimorphism’.
Point 4: I do however not see a way to include the first study, since its scope and results seem not to supplement the other two studies. Since it is without doubt an interesting manuscript by itself, I recommend to publish it separately.
Response 4: In this article, we would like to review/discuss fungal dimorphism in three different aspects: physiology (i.e. what can trigger dimorphic transition), reproductive biology (i.e. mating and ploidy status that affect dimorphism) and molecular genetics (i.e. signaling pathways and downstream regulation for dimorphic transition) using U. maydis as a primary model. Although these three sections seem independent, they are combined to explain the same theme, the biology of dimorphic fungi. There are recent reviews mentioning molecular pathways contributing to dimorphism in U. maydis (e.g. Vollmeister et al. 2012. FEMS Microbiology review 36: 59-77, Martinez-Soto et al. 2020 Microorganisms 8:1072), so we would like to extend the content of our article to cover other aspects of fungal dimorphism, as well as other known dimorphic fungi. Thus, we would like to retain these three sections in our article. We also explicitly write at the last paragraph of the introduction (line 74 – 75), mentioning that we are going to review/discuss fungal dimorphism in three different areas.
Point 5: I also have a few more detailed comments and suggestions, in particular to chapter 2:
line 54ff: I am not certain that dimorphic basidiomycetes are understudied on behalf of low economic or medicinal impact, and also do not think Ustilago is the best studied of these. What about Cryptococcus, which is a human pathogen, at least similarly well studied and also dimorphic when diploid?
Response 5: We acknowledge that Cryptococcus is another group of dimorphic fungi that are medically important. However, we argue this statement for a couple of reasons. First, despite being extensively studied, most research focuses on pathogenesis and interactions with host immune cells. Research in Cryptococcus dimorphism is primarily oriented towards the mating system and reproductive biology. Historically, the discovery of mating genes in Cryptococcus (Lengeler et al. 2002: Eukaryotic Cell 1(5) 704-718; Hull et al. 2005: Eukaryotic Cell 4(3):526-535) follows the ones in U. maydis (Bölker et al. 1992: Cell 68:441-450; Kronstad et al. 1990: 1384-1395), so does other genes like Crk1 (Garrido et al. 2004: Genes Dev 18: 3117-3130 in U.maydis and Liu and Shen 2011:48(3) 225-240 in C. neoformans). In addition, the knowledge in upstream signaling molecules that trigger dimorphic transition in Cryptococcus, rather than pheromone signaling, remains scarce. Other downstream effectors for hyphal growth like cell-end markers and polar-specific chitin synthases have never been reported in Cryptococcus, but these are well-characterized in U. maydis. The other reason is based on the purpose of this article. This article is dedicated for the special issue “Smut Fungi” of the Journal of Fungi. Thus, we primarily focus on knowledge from the smut model species U. maydis. However, we appreciate the reviewer’s comment. We mention Cryptococcus when discussing the role of mating and dimorphism (see a below response for line 207ff).
Point 6: line 121ff: Ustilago is not the only well studied plant pathogenic dimorph. For example, there are reports on the dimorphism trigger of the plant pathogens Ceratocystis ulmi (Naruzawa et al. 2016: Botany 94:31-39; Nickerson et al. 2006: Appl. Environ. Microbiol. 72: 3805-3813), Mycosphaerella graminicola (Mehrabi et al. 2007: Mol. Plant Microbe In. 19: 1262-1269) and even something on Taphrina (Svetaz et al. 2017: Plant Cell Environ. 40: 1456-1473).
Response 6: We define “well studied” by the fact that knowledge in fungal dimorphism covers not only physiological aspects (i.e. what can trigger dimorphic transition), but also mechanistic aspects (i.e. how signaling and response leads to dimorphic transition) and molecular genetic aspects (i.e. which genes/molecular players involved in the process). The abovementioned species lack knowledge in one of these aspects. However, according to the context in the specified line, we add more information about other dimorphic plant pathogens. We revise this section (line 121ff) as follows: “In the well-studied dimorphic plant pathogen U. maydis, acid pH, lipid and hydrophobicity have been demonstrated to promote filamentous growth. Linoleic acid can also induce hyphal growth in the Dutch-elm disease pathogen O. novo-ulmi. These cues are commonly found on the plant host surface, suggesting that the pathogens perceive the host for dimorphic transition. The peach leaf curl pathogen Taphrina deformans has filamentous growth only when it colonizes a host, possibly after perceiving an unknown signal.”
Point 7: line 133ff: Fatty acids are also involved in dimorphism of Ceratocystis (Naruzawa et al. 2016: Botany 94:31-39).
Response 7: We appreciate the reviewer’s comment, and we incorporate this information in Table 1 and the section of line 121ff (see an abovementioned response). We also did mention this in the same paragraph (see line 143-144 of the original version).
Point 8: line 161f: The growth of these three species is monomorphic only as far as your treatments could not trigger dimorphism. This does not imply that there is no dimorphism trigger at all.
Response 8: As we mention in our discussion that the dimorphism is not all-or-none phenotype (see Figure 2). We calculate the proportion of filamentous cells, and the three species have either all yeast cells or all filamentous cells for all treatments. According to this, it is convincing that these three species are monomorphic fungi.
Point 9: line 207ff: MAT seems also involved in Cryptococcus dimorphism (Wickes 1996: Proc. Natl. Acad. Sci. USA: 93: 7327-7331), and probably also in Tremella, since only their dikaryons are filamentous (and also parasitic).
Response 9: This article primarily focuses on U. maydis. However, we add information of Tremella in Table 1 that mating contributes to dimorphic transition. Also, we suggest readers to read a recent review on the mating system of Cryptococcus (Zhao, Y. et al. 2019. Life cycle of Cryptococcus neoformans. Annu. Rev. Microbiol. 73: 17–42). We edit the particular lines (207ff) as follows “Although quorum-sensing is poorly known in basidiomycetous fungi, mating is associated with dimorphism in many species. Hyphal growth is often found in the post-mating dikaryotic stage of dimorphic fungi in Pucciniomycotina, Ustilaginomycotina and Tremellomycetes. These include two model species: the human pathogenic fungus Cryptococcus neoformans and the corn smut fungus U. maydis. Although this article will focus on U. maydis, the mating system of C. neoformans and its role in filamentous growth can be found elsewhere.”
Point 10: line 545ff: I do not think you mean the hyphal form is "used" to invade host cells, since that implies a user. Maybe it is just able to enter them.
Response 10: We fix a particular sentence as “…, while the hyphal form can penetrate host tissues and evade host immunity”.
Point 11: References and figures are fine with me.
Table 2: I recommend to put table 2 to supplementary material. It disrupts the reading flow, and table 3 contains enough information to follow the text.
Response 11: We agree with the reviewer’s comment and put Table 2 to a supplementary file as Table S1.
Point 12: Table 3: Please indicate in the table legend what the blackened cells stand for.
Response 12: We add details in the legend as “White and black cell colors indicate ortholog presence and absence, respectively.”.
Reviewer 2 Report
The two authors summarize the current knowledge on fungal dimorphism across Asco-, Basidiomycetes and Mycuromycota. Fungal dimorphism is a phenomenon involving multiple protein players and many indirect effects. It even lacks clear definitions. The authors compiled a comprehensive review about the genetic basis including both direct and indirect players involved in regulating fungal dimorphism. Using the well-studied Basidiomycete model Ustilago maydis as a blueprint, they compare current knowledge on the different dimorphic fungi. They also comment on the potential biological background of fungal dimorphism which also varies strongly dependent on the different fungal lifestyles. The study is very specialized but well suited for the special issue on smut fungi. Minor points need to be addressed prior to publication (see comments).
Comments:
- Figure 1: it would be helpful to have close up pictures (for example as little inlays) of the yeast cells as they are too small in the overview to allow for a judgement of the reader. For some fungi I was not even able to really tell what the yeast morphology is like
Figure 1: why in some pictures the fungal cells look blueish? Is this only due to the use of phase contrast and no staining has been applied?
- Figure 2: Please explain a, b, c labelling in the figure.
- Lines 223/224: Please indicate that this is not a natural phenomenon but a genetic modification with artificial assembly of compatible mating genes.
- Figure 3: please correct pheromone in both cases in the arrow explanation legend
- Line 374: The authors might want to include results from Lanver et al.
- Lines 492 ff: Khd4 is likely involved in post-transcriptional regulation and not in long-distance transport, please adapt
https://www.ncbi.nlm.nih.gov/pmc/articles/PMC2779690/
- Table 3: please indicate what black /white colours in the respective table boxes mean (I guess it means absence/presence of a homolog). How was the presence of a gene defined? Whats the threshold of homology?Was there only one homolog present for each gene?
- Lines 537ff. I would have placed this paragraph to the top of the article.
Author Response
Point-by-point author response to reviewers
Reviewer 2
Comments and Suggestions for Authors
The two authors summarize the current knowledge on fungal dimorphism across Asco-, Basidiomycetes and Mycuromycota. Fungal dimorphism is a phenomenon involving multiple protein players and many indirect effects. It even lacks clear definitions. The authors compiled a comprehensive review about the genetic basis including both direct and indirect players involved in regulating fungal dimorphism. Using the well-studied Basidiomycete model Ustilago maydis as a blueprint, they compare current knowledge on the different dimorphic fungi. They also comment on the potential biological background of fungal dimorphism which also varies strongly dependent on the different fungal lifestyles. The study is very specialized but well suited for the special issue on smut fungi. Minor points need to be addressed prior to publication (see comments).
Comments:
Point 1: Figure 1: it would be helpful to have close up pictures (for example as little inlays) of the yeast cells as they are too small in the overview to allow for a judgement of the reader. For some fungi I was not even able to really tell what the yeast morphology is like
Response 1: We address this issue by adding inlay photos of yeast/filamentous cells in Figure 1. We also add the following sentence in Figure 1 legend: “An inlay photomicrograph in each cell is magnified up to 1.5 times to better visualize cell morphology.”.
Point 2: Figure 1: why in some pictures the fungal cells look blueish? Is this only due to the use of phase contrast and no staining has been applied?
Response 2: This is only due to the use of phase contrast. No staining has been applied in our experiments.
Point 3: Figure 2: Please explain a, b, c labelling in the figure.
Response 3: We revise the legend as “Letters indicate statistical groups inferred from the Tukey HSD post-hoc test, with ‘a’ as the highest value, ‘b’ as an intermediate value and ‘c’ as the lowest value.”
Point 4: Lines 223/224: Please indicate that this is not a natural phenomenon but a genetic modification with artificial assembly of compatible mating genes.
Response 4: We revise these lines as “Solopathogenic strains can originate either from diploid strains [70,71], or via genetic modification to artificially generate a single haploid strain with compatible bE and bW genes that can form a self-heterodimer.”
Point 5: Figure 3: please correct pheromone in both cases in the arrow explanation legend
Response 5: We thank the reviewer for noticing a typographical error. We correct it as suggested.
Point 6: Line 374: The authors might want to include results from Lanver et al.
Response 6: We add a sentence in a particular line as “Interestingly, the osmotic sensor protein Sho1 can directly interact with Kpp6 to trigger appressorium formation on a hydrophobic surface, but whether Sho1 can interact with other components in the MAPK cascade remains unknown.”.
Point 7: Lines 492 ff: Khd4 is likely involved in post-transcriptional regulation and not in long-distance transport, please adapt
https://www.ncbi.nlm.nih.gov/pmc/articles/PMC2779690/
Response 7: We thank the reviewer’s comment to improve the accuracy of the content. We revise and add the sentences as follows: “Finally, the RNA-binding proteins Khd4 and Rrm4 have been demonstrated to control regular filamentous growth. Rrm4 functions in long-distance mRNA transport so that transcripts involved in polarized growth can be expressed at a precise location. Khd4 recognizes a certain sequence motif in RNA, suggesting as role in post-transcriptional regulation.”
Point 8: Table 3: please indicate what black /white colours in the respective table boxes mean (I guess it means absence/presence of a homolog). How was the presence of a gene defined? Whats the threshold of homology? Was there only one homolog present for each gene?
Response 8: We add a sentence in the legend as “White and black cell colors indicate ortholog presence and absence, respectively.” The gene presence/absence was determined by the default parameters from OrthoFinder 2.1.2 (Emms, D.M.; Kelly, S. OrthoFinder : solving fundamental biases in whole genome comparisons dramatically improves orthogroup inference accuracy. Genome Biol. 2015, 16, 157, doi:10.1186/s13059-015-0721-2.). The OrthoFinder algorithm starts from determining homologs through all species blast search with e-value cutoff of 0.001. Then, the putative homologs were run for gene tree and species construction to determine orthology/paralogy. In some cases, there are more than one homolog in some species. We further inspect gene tree phylogeny generated from OrthoFinder to determine the gene homolog by monophyletic relationships.
Point 9: Lines 537ff. I would have placed this paragraph to the top of the article.
Response 9: We prefer keeping this towards the end of the article to provide broader ideas to the audience before the conclusion and perspectives, and not to distract the audience from the major contents of the article.
Round 2
Reviewer 1 Report
Point 1: The manuscript by T Kijpornjongpan and MC Aime aims to review the dimorphism of Ustilago maydis in comparison with non-model dimorphic fungi. While this is partly achieved, it does consist in fact of three studies which are almost independent of each other. Also, not all three fit into the proposed scope of this manuscript as review paper. The first study is purely experimental, analyzing the efficiency of different triggers for dimorphism in several dimorphic Ustilaginales strains, and showing that in most cases the triggers only incompletely shift the ratio between yeast and hyphal cells. The second study is also not a review, but a comparative genome organization analysis of the MAT genes in several Ustilaginales, concluding that there seems no pattern related to dimorphism. The third is indeed mainly a review, namely on the signaling pathways with emphasis on their relevance for triggering the dimorphism switch in Ustilaginales, but is also features genome comparison.
Response 1: Incorporating a few pieces of analyses to enhance discussion is not uncommon in recent review articles. See examples in Bakkeren et al. 2008. Fungal Genetics and Biology 45: S15 – S21; Riquelme and Bartnicki-Garcia 2008. Fungal Biology Reviews 22: 56-70; Kijpornyongpan et al. 2019. FEMS Yeast Research 19(2): foy125; Martinez-Soto et al. 2020 Microorganisms 8:1072; Libkind et al. 2020. FEMS Yeast Research 20: foaa042. However, most of them have parts of experiments/analyses embedded in the review, such as in a figure legend, and do not explicitly elaborate in the Materials and Methods section. We would like to explain what we have done in the Materials and Methods section to make sure other researchers can reproduce our work. Our article attempts to introduce each aspect of dimorphism by reviewing the literature, and then top up with our data to expand the border of research areas. Thus, our article may be considered as a hybrid between a research article and a review, but findings from the experiments/analyses are all used to enhance the discussion of the knowledge in the field.
--> It is certainly important to include a Materials and Methods section when incorporating original research into any kind of article. Still, the mention of other articles mixing reviews and original research does not convince me personally that this would be optimal for the present manuscript, or a desirable trend at all. I do believe this work would greatly benefit from a reorganization of the kind I suggested previously. Yet, I can see that the authors wish to proceed as they started, and also that the editor seems fine with the organization of the manuscript, so I will not stand in the way of its publication in the present form.
Point 2: The promise of the title is only partly fulfilled. The manuscript does quite some comparisons, but a discussion on the applicability of results obtained from the model for non-models, the degree of fitting for model species vs. non-model species, and on the validity of Ustilago as model species (e.g. for dimorphic fungi) is nowhere to be found. Perhaps the title might be rephrased to avoid disappointment in readers expecting these issues to be discussed.
Response 2: We acknowledge the reviewer’s comment and rename the title as “Investigating the smuts: common cues, signaling pathways, and the role of MAT in dimorphic switching and pathogenesis”.
--> This title fits way better, thanks a lot for considering my suggestion. Most of my previous concerns (i.e. all the "but Cryptococcus..." comments) were based on the fact that the previous title seemed to aim on the validity of Ustilago as model species for dimorphic basidiomycetes, which would have necessitated a more comparative discussion in my opinion.
Point 3: In my opinion, the first and third study are generally really good manuscripts in itself, while the second might need some reframing. There the authors seem surprised at the lack of patterns explaining the different growth forms, and thus conclude that comparing the MAT gene organization does not help to understand dimorphism. I recommend to frame it differently. It is in no way surprising that organisms do not completely restructure sexual reproduction as one of their core functions for a shift of growth form. This is especially valid if, as it seems, the potential for dimorphism is present in all Ustilaginales, if not all basal basidiomycetes as seen with the Tremellomycetes.
Instead, I would assume that a mechanistic linkage between sexual development and growth form has evolved several times independently, and perhaps also been lost repeatedly, in accordance with the life strategy of the respective species. Finding out where and how this linkage occurred might then indeed be studied by checking overlap in the respective signaling pathways. If done in this way, the second and third study might be combined to a well readable single manuscript.
Response 3: Fungal dimorphism is typically found in early-diverging basidiomycetes including Ustilaginomycotina, Pucciniomycotina and Tremellomycetes. These dimorphic fungi normally have hyphal growth during a dikaryotic stage after mating (Boekhout et al. 2011 in The Yeasts: A taxonomic study Vol III 5th edition. Chapter 100). Therefore, it is convincing to assume that sexual development is coupled with their dimorphic transition. One hypothesis is that the mating genes activate downstream genes involved in dimorphic transition from yeast to filamentous growth, as demonstrated in U. maydis. According to our comparative analyses (Table 3 in the original version and Figure 3), the mating genes as well as other upstream signaling pathways (like the MAPK pathway, activated by pheromone recognition) are well conserved in Ustilaginomycotina species, no matter what type of growth forms they have. Therefore, we conclude that these presence/absence patterns alone could not explain growth forms in studied fungal species and there must be other unknown molecular players, potentially in downstream pathways, that contribute to the coupling of sexual development and fungal dimorphism. As the knowledge in this area is scarce, it is too cumbersome to conduct a study as requested by the reviewer.
Even though our study could not provide any evidence how mating process is coupled with dimorphic transition, we would like to provide an update on the mating system of U. maydis and allies as previous studies were published around 10 years ago (Bakkeren et al. 2008. Fungal Genetics and Biology 45: S15 – S21; Kellner et al. 2011. PLOS Genetics 7(12): e1002436). Incorporating additional species that do not belong to Ustilaginales can provide better understanding how the mating loci are conserved across the subphylum. We also mention about the coupling of mating and dimorphic transition in the first and the last paragraphs of the section 3 ‘Mating and fungal dimorphism’.
--> I acknowledge the response of the authors. Indeed many dimorphic fungi switch to hyphal after dikaryotization. Yet the triggers are not restricted to this, and to me the phylogenetic pattern of that trait does not convincingly hint towards a single origin of dimorphism in basidiomycetes. Nevertheless I accept that thoroughly analyzing this topic is out of frame for this manuscript, and updating the published mating system for Ustilago is certainly also an achievement.
Point 4: I do however not see a way to include the first study, since its scope and results seem not to supplement the other two studies. Since it is without doubt an interesting manuscript by itself, I recommend to publish it separately.
Response 4: In this article, we would like to review/discuss fungal dimorphism in three different aspects: physiology (i.e. what can trigger dimorphic transition), reproductive biology (i.e. mating and ploidy status that affect dimorphism) and molecular genetics (i.e. signaling pathways and downstream regulation for dimorphic transition) using U. maydis as a primary model. Although these three sections seem independent, they are combined to explain the same theme, the biology of dimorphic fungi. There are recent reviews mentioning molecular pathways contributing to dimorphism in U. maydis (e.g. Vollmeister et al. 2012. FEMS Microbiology review 36: 59-77, Martinez-Soto et al. 2020 Microorganisms 8:1072), so we would like to extend the content of our article to cover other aspects of fungal dimorphism, as well as other known dimorphic fungi. Thus, we would like to retain these three sections in our article. We also explicitly write at the last paragraph of the introduction (line 74 – 75), mentioning that we are going to review/discuss fungal dimorphism in three different areas.
--> As written in my comment to Response 1, I believe separating the chapters into independent manuscripts would benefit all of them. Yet I accept the authors decision to keep them bundled.
Point 5: I also have a few more detailed comments and suggestions, in particular to chapter 2:
line 54ff: I am not certain that dimorphic basidiomycetes are understudied on behalf of low economic or medicinal impact, and also do not think Ustilago is the best studied of these. What about Cryptococcus, which is a human pathogen, at least similarly well studied and also dimorphic when diploid?
Response 5: We acknowledge that Cryptococcus is another group of dimorphic fungi that are medically important. However, we argue this statement for a couple of reasons. First, despite being extensively studied, most research focuses on pathogenesis and interactions with host immune cells. Research in Cryptococcus dimorphism is primarily oriented towards the mating system and reproductive biology. Historically, the discovery of mating genes in Cryptococcus (Lengeler et al. 2002: Eukaryotic Cell 1(5) 704-718; Hull et al. 2005: Eukaryotic Cell 4(3):526-535) follows the ones in U. maydis (Bölker et al. 1992: Cell 68:441-450; Kronstad et al. 1990: 1384-1395), so does other genes like Crk1 (Garrido et al. 2004: Genes Dev 18: 3117-3130 in U.maydis and Liu and Shen 2011:48(3) 225-240 in C. neoformans). In addition, the knowledge in upstream signaling molecules that trigger dimorphic transition in Cryptococcus, rather than pheromone signaling, remains scarce. Other downstream effectors for hyphal growth like cell-end markers and polar-specific chitin synthases have never been reported in Cryptococcus, but these are well-characterized in U. maydis. The other reason is based on the purpose of this article. This article is dedicated for the special issue “Smut Fungi” of the Journal of Fungi. Thus, we primarily focus on knowledge from the smut model species U. maydis. However, we appreciate the reviewer’s comment. We mention Cryptococcus when discussing the role of mating and dimorphism (see a below response for line 207ff).
--> While the authors are certainly correct in their answer, they just emphasize my point. It is true that Ustilago is, of all dimorphic basidiomycetes, best studied in terms of mating and dimorphism, and thus a fitting object to study in this respect. But the reasons for that lie certainly not in the lack of economic or medicinal impact with either Cryptococcus or Ustilago, and there are other aspects where there is is far more known about Cryptococcus. Nevertheless, I am fine if the sentence is kept as it is, as the new title does not focus anymore on comparison of model organisms for dimorphic basidiomycete fungi.
Point 6: line 121ff: Ustilago is not the only well studied plant pathogenic dimorph. For example, there are reports on the dimorphism trigger of the plant pathogens Ceratocystis ulmi (Naruzawa et al. 2016: Botany 94:31-39; Nickerson et al. 2006: Appl. Environ. Microbiol. 72: 3805-3813), Mycosphaerella graminicola (Mehrabi et al. 2007: Mol. Plant Microbe In. 19: 1262-1269) and even something on Taphrina (Svetaz et al. 2017: Plant Cell Environ. 40: 1456-1473).
Response 6: We define “well studied” by the fact that knowledge in fungal dimorphism covers not only physiological aspects (i.e. what can trigger dimorphic transition), but also mechanistic aspects (i.e. how signaling and response leads to dimorphic transition) and molecular genetic aspects (i.e. which genes/molecular players involved in the process). The abovementioned species lack knowledge in one of these aspects. However, according to the context in the specified line, we add more information about other dimorphic plant pathogens. We revise this section (line 121ff) as follows: “In the well-studied dimorphic plant pathogen U. maydis, acid pH, lipid and hydrophobicity have been demonstrated to promote filamentous growth. Linoleic acid can also induce hyphal growth in the Dutch-elm disease pathogen O. novo-ulmi. These cues are commonly found on the plant host surface, suggesting that the pathogens perceive the host for dimorphic transition. The peach leaf curl pathogen Taphrina deformans has filamentous growth only when it colonizes a host, possibly after perceiving an unknown signal.”
--> I acknowledge the revision of this section and thank the authors for considering my comment. Of course Ustilago is best studied among the mentioned species. My point here was mainly to avoid overgeneralization considering the previous title of the manuscript.
Point 7: line 133ff: Fatty acids are also involved in dimorphism of Ceratocystis (Naruzawa et al. 2016: Botany 94:31-39).
Response 7: We appreciate the reviewer’s comment, and we incorporate this information in Table 1 and the section of line 121ff (see an abovementioned response). We also did mention this in the same paragraph (see line 143-144 of the original version).
--> I thank the authors for considering my suggestion.
Point 8: line 161f: The growth of these three species is monomorphic only as far as your treatments could not trigger dimorphism. This does not imply that there is no dimorphism trigger at all.
Response 8: As we mention in our discussion that the dimorphism is not all-or-none phenotype (see Figure 2). We calculate the proportion of filamentous cells, and the three species have either all yeast cells or all filamentous cells for all treatments. According to this, it is convincing that these three species are monomorphic fungi.
--> I must say I am not completely convinced by this argument. Showing the majority of the studied species have an incomplete change of phenotype after triggering does not imply that this is necessarily the case for other species. There might e.g. be a time lag, a diffusion of signal on agar cultures, or as previously written an unknown trigger involved. Yet I agree that for the purpose of this study, they might be considered monomorphic under the conditions checked here.
Point 9: line 207ff: MAT seems also involved in Cryptococcus dimorphism (Wickes 1996: Proc. Natl. Acad. Sci. USA: 93: 7327-7331), and probably also in Tremella, since only their dikaryons are filamentous (and also parasitic).
Response 9: This article primarily focuses on U. maydis. However, we add information of Tremella in Table 1 that mating contributes to dimorphic transition. Also, we suggest readers to read a recent review on the mating system of Cryptococcus (Zhao, Y. et al. 2019. Life cycle of Cryptococcus neoformans. Annu. Rev. Microbiol. 73: 17–42). We edit the particular lines (207ff) as follows “Although quorum-sensing is poorly known in basidiomycetous fungi, mating is associated with dimorphism in many species. Hyphal growth is often found in the post-mating dikaryotic stage of dimorphic fungi in Pucciniomycotina, Ustilaginomycotina and Tremellomycetes. These include two model species: the human pathogenic fungus Cryptococcus neoformans and the corn smut fungus U. maydis. Although this article will focus on U. maydis, the mating system of C. neoformans and its role in filamentous growth can be found elsewhere.”
--> I agree that an Ustilago manuscript must focus on Ustilago, especially in a special issue focused on smut fungi. And with the new title, its role as model species for dimorphic basidiomycetes seems no more the main focus. So I thank the authors for considering my comment and for the editing the section.
Point 10: line 545ff: I do not think you mean the hyphal form is "used" to invade host cells, since that implies a user. Maybe it is just able to enter them.
Response 10: We fix a particular sentence as “…, while the hyphal form can penetrate host tissues and evade host immunity”.
--> I thank the authors for considering my suggestion.
Point 11: References and figures are fine with me.
Table 2: I recommend to put table 2 to supplementary material. It disrupts the reading flow, and table 3 contains enough information to follow the text.
Response 11: We agree with the reviewer’s comment and put Table 2 to a supplementary file as Table S1.
--> I thank the authors for considering my suggestion.
Point 12: Table 3: Please indicate in the table legend what the blackened cells stand for.
Response 12: We add details in the legend as “White and black cell colors indicate ortholog presence and absence, respectively.”.
--> I thank the authors for considering my suggestion.